# Stereochemical and Biosynthetic Rationalisation of the Tropolone Sesquiterpenoids

**DOI:** 10.3390/jof8090929

**Published:** 2022-08-31

**Authors:** Lei Li, Russell J. Cox

**Affiliations:** 1Institute for Organic Chemistry and BMWZ, Leibniz University of Hannover, Schneiderberg 38, 30167 Hannover, Germany; 2Department of Chemistry, BioDiscovery Institute, University of North Texas, 1155 Union Circle, Denton, TX 76201, USA

**Keywords:** tropolone, meroterpenoid, sesquiterpenoid, stereochemical revision, biosynthesis

## Abstract

This review summarises the known structures, biological activities, and biosynthetic pathways of the tropolone sesquiterpenoid family of fungal secondary metabolites. Synthesis of this knowledge allows likely structural and stereochemical misassignments to be revised and shows how the compounds can be divided into three main biosynthetic classes based on the stereochemistry of key biosynthetic steps.

## 1. Introduction

Tropolone sesquiterpenoids are a class of fungal secondary metabolites that contain the structural motif of a core 11-membered macrocycle derived from humulene, which is connected to one or two polyketide-derived tropolones via dihydropyran rings. They possess diverse and potent bioactivities. Recent advances in their total synthesis, biosynthesis, and bioengineering have revealed new aspects of their structures. Biosynthetic studies, in particular, have shown that they are formed via a key hetero-Diels–Alder (hDA) reaction between a tropolone quinomethide derived from a polyketide pathway and the sesquiterpene humulene. Key family members include: the bistropolones pycnidione **1** [1,2] and eupenifeldin **2** [3]; the mono-tropolones epolone B **3** [2,4] and neosetophomone B **4** [5]; and the xenovulenes, such as **5** [6,7], in which the polyketide-derived moiety has been extensively oxidatively modified (Figure 1).

The tropolone sesquiterpenoids have been isolated from diverse fungi, some of which are themselves poorly characterised, over a wide time period beginning in 1993 [1]. There has therefore been a variety of naming conventions used. It has also proven difficult in a number of cases to accurately derive the correct relative and absolute stereochemistries in the 11-membered humulene-derived rings. However, recent synthetic work by Sarlah and co-workers [8] has made a significant contribution to clarifying configurational ambiguities in this area, and new methods, including vibrational circular dichroism (VCD) and electronic circular dichroism (ECD) coupled to single-point ωB97X/def2-TZVP calculation, have revealed the likely absolute stereochemistries. Until recently, biosynthetic hypotheses about the origin of these compounds have relied on classical isotopic feeding experiments [6], but new understanding from genetic and biochemical perspectives now helps to rationalise the biosynthetic pathways and the structures.

Here, we wish to bring together all current knowledge on tropolone sesquiterpenoids with the aim of understanding and rationalising the structures (stereochemistries in particular) and biosynthesis.

## 2. Nomenclature and Numbering

For the most part, compounds are named after the producing organisms, or in the case of xenovulenes after the investigating company, or after their biological properties (e.g. epolones induce erythropoietin (EPO) production) [4]. Thus, names of structurally related compounds are often unrelated. However, Singh and co-workers introduced the term ‘nor’ as in noreupenifeldin **6** to refer to ring-contracted congeners (i.e., lacking one carbon) [9].

Thus, eupenifeldin **2** and noreupenifeldin **6** are tropolone and phenol congeners. However, the equivalent pair of compounds in the pycnidione series are pycnidione **1** and epolone A **7** [2]. Likewise, neosetophomone B **4** [5] and pughiinin A **8** [10] are related by a single ring contraction of the western tropolone (Figure 2). Further oxidative rearrangements of the aromatic rings lead to compounds with a wide variety of trivial names, e.g. phomanolides [11,12,13] and phomeroids [14]. Lettering, e.g. xenovulene A **5** and xenovulene **B** 9 (Figure 2), usually only reflects the order of reporting of related compounds and has no biosynthetic relevance.

A wide variety of numbering schemes for the tropolone sesquiterpenoids have been used in the literature. The first compound to be reported, pycnidione **1** in 1993 [1], used a scheme starting at the first CH in the eastern tropolone and proceeding anticlockwise around the pentacycle periphery. Eupenifeldin **2** (also 1993) [3] was numbered 1–8 in the western ring, 1′ to 8′ in the eastern ring, and 1″ to 19″ for the fifteen humulene carbons, the two pyran oxygens, and the two tropolone-derived pyran methylenes. Later (1995), xenovulene A **5** [6] was numbered 1–11 for the humulene backbone, 12–18 for the polyketide-derived carbons, and 19–22 for the peripheral methyls. Report of the discovery of the epolones **3** and **7** (1998) introduced another different numbering scheme and also renumbered pycnidione **1** at the same time [4].

The next compound to be discovered, pughiinin A **8** (2002) [10], used yet another different scheme, while ramiferin **10** (2008) [15] used a numbering system similar to that of pycnidione **1**. Thus, a wide variety of different numbering systems have been used to date.

Here, we propose a standardised biosynthetic numbering system since the biosynthetic relationships are now clear (Figure 3). Thus, humulene-derived carbons are numbered 1–15 according to the numbering of the humulene precursor farnesyl diphosphate **29** (FPP). In this proposed scheme, the geminal dimethyl carbon is C-11, and the methylene to the right is C-1. Methyls are numbered 12–15 anticlockwise, with the C-12 methyl defined as the *pro-R* substituent. The western tropolone is then numbered 1′–9′ according to its precursor polyketide with C-1′ marking the head of the chain and C-9′ the tail; the eastern tropolone (if present) is numbered 1″–9″ with the same logic.

## 3. Biological Activities

In 1993, pycnidione **1** was first discovered as a selective stromelysin inhibitor, with potential applications in the treatment of arthritis [1]. After that, Cai and co-workers reported that pycnidione **1** induced erythropoietin gene expression fivefold at a concentration of 1.6 μM [4]. Recombinant human erythropoietin is used in the treatment of patients with anaemia caused by chronic renal failure, cancer chemotherapy, and a variety of other disease states [4].

Novel bioactivities for pycnidione **1** are continually reported and include antimicrobial [2], anticancer [16,17], and anthelmintic activities [18,19]. In 2021, pycnidione **1** was patented as an active principle of anti-dandruff shampoo [20]. In the course of screening for small molecule modulators of erythropoietin gene expression, epolone A **7** and epolone B **3** were also discovered to induce erythropoietin gene expression five-fold in low μM concentrations [4,21].

Eupenifeldin **2** possesses several different biological activities, such as anticancer [3,22], antiplasmodial [10], antifungal [15], and in vitro anthelmintic activities [9]. In addition, a few tropolone sesquiterpenoids identified together with eupenifeldin **2** were also reported as bioactive compounds. Pughiinin A **8**, isolated from the seed fungus *Kionochaeta pughii* BCC 3878, exhibits in vitro antiplasmodial activity against *Plasmodium falciparum* K1 with IC_50_ values of 2.4 μg/mL [10]. Later, ramiferin **10**, isolated from the fungus *Kionochaeta ramifera* BCC 7585, was found to exhibit activities against *Plasmodium falciparum K1* with an IC_50_ value of 6.3 μM, and is toxic against three cancer cell lines (BC, KB, and NCI-H187) with respective IC_50_ values of 9.1, 12.6, and 13.0 μM [15]. Phomanolides A **11**, D **12**, and F **13**, isolated from a *Phoma* sp. stain, exhibit cytotoxic effects toward human glioma cell lines (Figure 2) [11,12]. During cytotoxic screening against four human cancer lines: SF-268 (human glioblastoma carcinoma), MCF-7 (breast cancer), HepG-2 (liver cancer), and A549 (lung cancer), phomeroid B **14** showed significant antiproliferative effects with IC_50_ values of 0.50–1.30 μM, while phomeroid A **15** exhibited moderate activities [14].

In addition, dehydroxyeupenifeldin **16** and noreupenifeldin B **17**, isolated from *Neosetophoma* sp., were observed to show cytotoxic effects toward a panel of cancer cell lines, including MDA−MB−231 (human breast cancer), OVCAR-3 and OVCAR-8 (human ovarian cancers), MSTO-211H (human mesothelioma), and LLC (murine lung cancer) [5].

A study on screening new inhibitors of the binding of flunitrazepam to the GABA benzodiazepine receptor led to the discovery of xenovulene A **5** from *Acremonium strictum*. It is an inhibitor of benzodiazepine binding to the γ-aminobutyric acid (GABA) receptor, with potential use as an anti-depressant with reduced addictive properties [6].

## 4. Biosynthesis

In all cases studied to date, the biosynthesis of tropolone sesquiterpenoids begins with the construction of a tetraketide tropolone. This follows the same early steps as known for the construction of the classical fungal tropolone stipitatic acid **27** (Figure 1A) [23]. In brief, a non-reducing polyketide synthase (TropA) constructs 3-methylorcinaldehyde **20**. This is then oxidised by an FAD-dependent monooxygenase (FMO, TropB) [24,25] to give the dearomatised nucleus **21**. Further oxidation by a non-heme iron dioxygenase (TropC) then gives the re-aromatised tropolone nucleus **22** (Figure 1A) [26].

In the eupenifeldin-family and pycnidione-family pathways, the tropolone aldehyde is reduced by a short-chain reductase–dehydrogenase (SDR) to give the corresponding primary alcohol **25** [27]. In contrast, in the case of the xenovulene series, the initial aldehyde is oxygenated by a cytochrome P450 monooxygenase (TropD) to give the cyclic acetal **23** [28].

In parallel, humulene is synthesised from FPP **29** by a non-canonical sesquiterpene cyclase [28]. Remarkably, a single mutation in the active site of this enzyme can select the synthesis of either *EEE* or *ZEE* humulene (Figure 1B). The next step is the hetero-Diels–Alder (hDA) reaction [29]. Evidence shows that in the eupenifeldin/pycnidione cases, the hDA enzyme dehydrates the tropolone alcohol **25** to form a transient quinomethide **26** prior to the stereospecific hDA reaction with the humulene dienophile to form a dihydropyran (Figure 1C) [27].

In the xenovulene case, it appears that the hemiacetal **23** is dehydrated to form an analogous quinomethide **24** that reacts with humulene in the same way [28]. All evidence to date suggests that the hDA reaction is stereospecific. In the cases of the pycnidione **1** and xenovulene A **5** pathways, the humulene partner is the all *E* diastereomer. However, in the eupenifeldin **2** pathway, the humulene is the *Z**EE* diastereomer. The *EEE* or *Z**EE* diastereomers of humulene are formed by highly similar, but also highly diastereoselective, sesquiterpene cyclases (Figure 1B). Although the sequences of these cyclases are highly diverged from all currently known terpene cyclases, they, in fact, have fairly standard structures and mechanisms [28].

Thus, three main families of tropolone sesquiterpenoids are defined by two biosynthetic steps: oxidative chemistry prior to hDA reaction in the case of the xenovulenes or reductive chemistry prior to hDA reaction in the case of the pycnidiones and eupenifeldins; and use of *EEE* humulene **28** in the cases of the pycnidiones and xenovulenes and *Z**EE* humulene **30** in the case of the eupenifeldins.

Dividing the compounds into these three main families based on early reaction selectivity allows all the known structures to be grouped, and for other observed transformations to be understood to happen later. Thus, one major family is constructed around eupenifeldin **2** (Figure 2), a second around pycnidione **1** (Figure 3), and a third around xenovulene A **5** (Figure 4). Biosynthetic considerations then allow the later steps to be rationalised.

In the cases of the eupenifeldin and pycnidione families, there is usually a second hDA reaction to build a tetrahydropyranotropolone on the eastern side (e.g. **2** and **7** themselves). Since these reactions are stereospecific, the *Z*-olefins of the eupenifeldin series give rise to *syn*-diastereomers at the new ring junction, while the *E*-olefins of the pycnidione series afford *anti* ring junctions at this position.

Following these well-investigated early steps, the pathways become more diverse. Oxygenation of the humulene can occur, such as hydroxylation at C-4. It is worth noting that all the known eupenifeldin series have a 4*S* configuration (but see Section 6), while the pycnidione series have a 4*R* configuration, indicating that this is a stereoselective hydroxylation reaction. However, the most common reactions appear to be oxidative ring contraction of the tropolones to form benzopyrans. This can occur at either (or both) of the eastern or western tropolones. Further oxidative ring contractions and rearrangements, thus-far always observed on the western side, give a very wide variety of pyrones and γ-lactones exemplified by the phomanolides and phomeroids in the eupenifeldin series (Figure 2), the epolones in the pycnidione family (Figure 3), and the later xenovulenes (Figure 4).

Some of the ring contractions are characterised. For example, in the case of the xenovulenes, two successive oxidative contractions are catalysed by FMO enzymes (AsL4 and AsL6, Figure 4) [28]. In *A. oryzae*, used for the heterologous production of some engineered compounds, an uncharacterised serendipitous endogenous enzyme can catalyse a similar transformation (e.g. **19** to **40**, Figure 3)—albeit with a likely different mechanism [30]. However, for the more heavily oxidised compounds such as **31**–**33** (Figure 2), the responsible genes and enzymes have not yet been discovered.

## 5. Biosynthetic Gene Clusters

Biosynthetic gene clusters (BGC) encoding the biosynthesis of stipitatic acid **27** [23], xenovulene A **5** [28], eupenifeldin **2** [27,30], and pycnidione **1** [30] are known (Figure 4). All BGCs contain genes encoding the core PKS (e.g. TropA), FMO (e.g. TropB), and NHI dioxygenase (e.g. TropC) that catalyse the construction of the tropolone nucleus **22**. The stipitatic acid and xenovulene clusters encode a cytochrome P450 monooxygenase (TropD) that performs oxygenation of the methyl and formation of stipitacetal **23** [23,28]. This gene is not present in the other BGCs that do not require this step. The xenovulene A **5**, eupenifeldin **2**, and pycnidione **1** BGCs encode the required humulene synthase (e.g. AsR6) [30]. This enzyme is significantly diverged from more well-studied terpene cyclases [28]. The xenovulene A **5**, eupenifeldin **2**, and pycnidione **1** BGCs also encode the hetero-Diels–Alderase (hDA, e.g. AsR5). The genes encoding these two functionalities are common to all the tropolone sesquiterpenoid BGCs.

Further common genes include *asL4* and *asL6* that are homologs of *eupR4*, *eup2R5*, and *pycR4* encoding FMOs most likely encoding enzymes involved in tropolone ring contractions. These activities have been confirmed in the cases of AsL4, AsL6, and Eup2R5 [28,30]. The *eup*, *eup2*, and *pyc* BGCs all contain a gene encoding an SDR (e.g. PycL2) required for reduction of stipitaldehyde **22** to stipitol **25**, and a gene encoding a cytochrome P450 monooxygenase (e.g. PycR5) that is responsible for the hydroxylation of the humulene moiety at C-4. The tropolone sesquiterpenoid BGCs appear to have common transporter genes (e.g. *asL7*), but the transcription factor genes in the xenovulene BGC are different to common genes (e.g. *eup2R4* and *eup2R2*) in the *eup* and *pyc* BGCs.

## 6. Stereochemical and Structural Revisions

Accurate determination of the relative and absolute stereochemistries of the tropolone sesquiterpenoids has not been simple, and until recently, several stereochemical inconsistencies were present in the literature. These are summarised in Figure 2, Figure 3 and Figure 4. Biosynthetic arguments are important in understanding these stereochemical relationships. The first stereocentres to be set are at C-6 and C-7 via the enzyme-catalysed hDA reaction. Since the hDA enzymes from the three families are highly similar, it is very likely that the C-6/C-7 *anti* ring junction should have the same absolute stereochemistry in all three series.

In the case of eupenifeldin **2**, X-ray analysis confirmed the relative stereochemistry [3,32]. VCD measurements by Oberlies and co-workers [5] and ECD measurements by Che and co-workers [12] were used to determine its absolute configuration (Figure 5). These data conclusively set the absolute stereochemistry of **2** as *opposite* to that drawn originally, and X-ray and MTPA analysis of its putative precursor neosetophomone B **4** confirm this conclusion.

Although compounds in this series have been isolated from various sources, it seems highly unlikely that BGCs have evolved that specify the synthesis of enantiomeric compounds. It is therefore likely that all compounds in the eupenifeldin series should be 6*S*, 7*S*. However, since enantiomeric pairs of natural products are known in other cases [33,34], this hypothesis cannot be confirmed without further detailed stereochemical investigations. Additionally, recent synthetic work by Sarlah and co-workers, in the case of pycnidione **1**, also confirms the 6*S*,7*S* ring junction stereochemistry (Figure 5) [8]. It is therefore surprising that pughiinin A **8** and ramiferin **10** that co-occur with eupenifeldin **2** in *Kionochaeta ramifera* are assigned as 6*R*,7*R*. Since, in both cases, the C-4 hydroxyl is also reported to be inverted relative to the C-4 hydroxyl of eupenifeldin **2**, it is likely (but subject to the caveat above) that pughiinin A **8** and ramiferin **10** should be reported as their respective enantiomers (revised structures shown in Figure 2).

In the pycnidione series (Figure 3), Sarlah’s recent synthetic work has proven the western *anti* ring junction to be 6*S*,7*S* as mentioned before. This showed that the originally reported structures (e.g. pycnidione **1** and epolone B **3**) were enantiomeric to the true structures. Even though the stereochemistry of the 4-hydroxyl cannot be determined as in the eupenifeldin series, the observed nOe measurements (H-4 and H-2) for pycnidione **1** support 4*R* stereochemistry [4]. This was supported by Sarlah’s synthesis of the enantiomeric (-)-epolone B that was definitively proven to be 4*S*,6*R*,7*R* [8]. Finally, X-ray studies of naturally occurring pycnidione **1** itself confirm the relative and absolute stereochemistries at the Eastern 2*S*,3*S* ring junction (Figure 5). Since all later structures have been assigned on the basis of the original enantiomeric structure, it seems likely that all of the pycnidione family should be reassigned accordingly, including epolones A **7**, B **3**, and C **18** [2], and dehydroxypycnidione **19** [2]. Proposed revised structures are shown in Figure 3.

Phomeroids A **15** and B **14** are reported to have an inverted secondary alcohol at C-4 (4*R*) relative to their proposed precursor eupenifeldin **2** (i.e. 4*S*). Since both compounds co-occur with eupenifeldin **2** in *Phomopsis tersa*, it is noteworthy that opposite absolute configurations have been determined at C-4. The absolute configuration of eupenifeldin **2** was determined by a combination of X-ray, VCD, and ECD methods, while the configurations of **14** and **15** were determined by comparison of experimental and TDDFT-calculated ECD. Related compounds phomanolides A-F (**11***–***13**, **32**, **33** and **35**) are all assigned as 4*S*, and all also co-occur with eupenifeldin **2**. NMR analysis of pycnidione **1** and eupenifeldin **2** shows that the C-4 stereochemistry correlates with the proton and carbon chemical shifts (Figure 5). In the 4*R* epimer, δ_H-4_ = 3.60 ppm and δ_C-4_ = 77.0 ppm, while in the 4*S* epimer, δ_H-4_ = 4.22 ppm and δ_C-4_ = 70.8 ppm. The observed δ_H-4_ = 4.07 ppm, δ_C-4_ = 71.5 ppm in phomeroid A **15** and δ_H-4_ = 4.15 ppm, δ_C-4_ = 70.7 ppm in phomeroid B **14** are consistent with 4*S* configuration. NMR analysis of epimers **36** and **37** (δ_H-4_ = 3.99 ppm, δ_C-4_ = 78.4 ppm in **36**; δ_H-4_ = 4.35 ppm, δ_C-4_ = 73.9 ppm in **37**), created in a synthetic biology project in *A. oryzae* [30], also follows the same chemical shift pattern. The relative configurations of **36** and **37** are confirmed by a series of 1D-NOE experiments.

Indeed, nOe data also support the hypothesis that **14** and **15** may be 4*S* configured. In eupenifeldin **2**, H-4 and H-14 are located 2.9 Å apart and would be expected to show an nOe correlation [3]. In ramiferin **10** that has the same overall skeleton as **2** (except tropolones are replaced by benzenes), a strong nOe correlation is indeed observed between H-4 and CH_3–_14 (Figure 5) [15]. The same nOe correlation is observed for the phomanolides A-F (**11***–***13**, **32**, **33** and **35**) that are also assigned as 4*S* [12]. Phomeroids A **15** and B **14** are also reported to display the same nOe correlation suggestive of the 4*S* configuration [14]. Thus, chemical shift and nOe NMR arguments suggest **14** and **15** are 4*S* configured, but comparison of experimental and TDDFT-calculated ECD for these compounds suggests the opposite.

However, NMR measurements alone are not sufficient to solve the conundrum of the proposed 4*R*-stereochemistry in **14** and **15**. If the 4-hydroxyls in **14** and **15** are actually 4*R*, they would have to be introduced independently, or possibly derived from an initial 4*S* alcohol that is oxidised to a ketone and re-reduced to the 4*R* epimer. This mechanism may also explain the observation of both 4-epimers in the cases of **36** and **37** where the 4-hydroxyl is introduced by the eupenifeldin Eup2R6 P450 oxygenase in *A. oryzae* (Figure 3). It may also be that the eupenifeldin Eup2R6 P450 that activates the 4-position is non-specific and produces both epimers—although this is rare. Further detailed biosynthetic and structural work will be required to shed light on this inconsistency.

The relative stereochemistry of the xenovulene series (Figure 4) was assigned on the basis of an X-ray structure, but the absolute stereochemistry has not been determined. Since the initial chirality is generated by the hDA enzyme that is highly similar to the pycnidione **1** and eupenifeldin **2** hDA enzymes, it seems highly likely that this should be 6*R*,7*S*—opposite to that originally proposed (Figure 5).

Finally, appreciation of the biosynthetic basis of production of the tropolone sesquiterpenoids allows an apparent anomaly to be resolved. Pittayakhajonwut and co-workers reported the isolation of pughiinin A **8** and its structure elucidation by X-ray crystallography, from the fungus *Kionochaeta pughii* BCC 3878 [10]. Pughiinin A **8** is a ring-contracted congener of neosetophomone B **4**, clearly derived from *Z**EE*-humulene **30**, and it therefore belongs to the eupenifeldin family (Figure 2). It was therefore surprising that pycnidione **1** was reported as a co-metabolite, as pycnidione **1** arises from *EEE* humulene **28**. However, comparison of the NMR data of the compound with those of the reported eupenifeldin and pycnidione spectra shows that it is, in fact, eupenifeldin **2** and therefore consistent with the idea that individual fungal species contain *either* eupenifeldin-type [4] or pycnidione-type BGCs [12].

## 7. Related Compounds

Recently reported meroterpenoids photeroids A **50** and B **51** from *Phomopsis tersa* have been speculated to be derived via an hDA reaction between a β-cadinene **48** and a polyketide-derived orthoquinomethide (Figure 5) [35]. Since this *Phomopsis* species is also known to create the phomeroid tropolone sesquiterpenoids [14] it could also be possible that a tropolone is a precursor in this case, giving a tropolone intermediate **49** that then undergoes one of the frequently observed ring contraction reactions. A simple isotopic feeding experiment as demonstrated previously would address this possibility [30]. The likely close relationship between the photeroid and phomeroid pathways in *P. tersa* is supported by the observation of analogous stereoselectivity of the putative hDA reactions.

Related compounds include the bisbenzopyran-fused compound lucidene **52** from the plant *Uvaria lucida* ssp. *lucida* [36] and the sterhirsutins A **53** and B **54** from the fungus *Stereum hirsutum* [37]. In the case of lucidene **52**, it appears that tropolones are unlikely to be biosynthetic precursors because of the highly reduced nature of the aromatic rings. The hirsutins clearly derive via condensation of *EEE* humulene and hirsutane sesquiterpenoids. An hDA route has been proposed for the biosynthesis in this case, but the observation of both stereoisomeric products **53** and **54** in this case suggests the condensation may be non-enzymatic. Such hDA reactions have been extensively reviewed elsewhere [38].

## 8. Conclusions

Elucidation of the biosynthesis of the tropolone sesquiterpenoids, coupled with recent advances in analytical and synthetic chemistry, has allowed a much fuller understanding of the construction of this family of bioactive fungal metabolites. The diversity within this family is generated firstly by the selective formation of either *EEE* or *Z**EE* humulene, and the stereospecific hDA reaction that forms the first pyran ring junction. Further diversity is incorporated by a frequent second hDA reaction to create the *bis*-tropolones. P450 oxygenation of the C-4 position appears to be performed early and both C-4 epimers are known. Finally, a very wide range of oxidative transformations of the tropolones has been observed, often yielding ring-contracted and rearranged products. Current knowledge in this area is restricted to the understanding of the ring-contraction steps involved in xenovulene biosynthesis, although in vitro and mechanistic details are still lacking.

In addition, absolute configurations of a few tropolone sesquiterpenoids have been revised. Within the pycnidione series, bistropolone sesquiterpenoids (e.g. pycnidione **1**) contain a 4*R* hydroxyl group and *anti*-relationship at the eastern dihydropyran ring junction, while a mono-tropolone sesquiterpenoid (e.g. epolone B **3**) bears a 4*R* hydroxyl group and 2*E*-alkene and lacks the eastern dihydropyran ring junction. The diastereomeric eupenifeldin-type maintain an identical trend among compounds but are differentiated from the former by 4*S* hydroxyl group, the *syn*-relationship at the eastern dihydropyran ring junction in bistropolone sesquiterpenoids (e.g. eupenifelin **2**), and 2*Z*-alkene in the mono-tropolone sesquiterpenoids (e.g. neosetophomone B **4**).

The tropolone sesquiterpenoids possess diverse and often potent bioactivities, and it is likely that further compounds within this class will be discovered with such properties. Rational creation of these compounds is also becoming possible via chemical synthesis or synthetic biology. It appears likely, therefore, that this class of metabolites will remain of high interest to organic chemists in years to come.

## Data Availability

Not applicable.

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
