# Peer review of "Stereochemical and Biosynthetic Rationalisation of the Tropolone Sesquiterpenoids"

_jof, 2022, doi:10.3390/jof8090929_

Round 1

Reviewer 1 Report

This review summarized the structures, biological activities and biosynthesis of topolone sesquiterpenoids, which is readable and instructive to a certain extent. But like many similar review papers, this paper still needs to emphasize the basis for the topic selection. In addition, for the level and positioning of this journal, only 35 citations are slightly insufficient. Therefore, it is more important for the authors to put forward constructive guiding ideology through the review.

Reviewer 2 Report

I found these natural products very interesting, with intriguing structures. Characterization of these tropolone sesquiterpenes is really challenging.

I think that spectroscopic information should be presented in the article, or as supplementary material, it would be useful for readers. 

Some other comments

Page 1

Title: 

Line 2: "Rationalization" instead of "Rationalisation"

Introduction 

Page 1

Line 23: Provide reference of total synthesis and biosynthesis, maybe these are the ones cited later on the text 

Line 28: Provide reference, I think is reference (1)

Line 33: I´d suggest replacement of "questions" for "configurational ambiguities"

Line 34: "high level calculation": A more specific description of the calculation method used should be provided 

Line 39: "tropolone" instead of "topolone" 

Page 2

Figure 1.  "tropolone" instead of "topolone"

Figure 2. The numbers of the structures doesn´t follow order of appeareance in text, I think that compounds of the same reference are grouped together.  

Line 48: "e.g. epolones induce erytropoietin [EPO] production. Provide reference

Line 50: According to IUPAC, the term nor- is used along the position, in this case it would be 6´(or 5´)-noreupenifeldin 

Page 3

Line 64: "anticlockwise" instead of "anticlockise" 

Line 93: I suggest "...was patented as an active principle of anti-dandruff shampoo" instead of "...was used as anti-dandruff composition in shampoo"  

Page 4 

Scheme 1 A. "tropolone" instead of "topolone" , it seems that the structure of ZEE-humulene 30 have a methyl in the C-2 position, but it should be in C-3

Page 5

Line 134 to 138. references 26-28 could be listed in order of appeareance

Line 141: replace alchohol for alcohol

Line 141-143. "hDA enzyme dehydrates the tropolone alchohol 25 to form a transient quinomethide 26 that reacts with the humulene dienophile to form a dihydropyran prior to the stereospecific hDA reaction" 

The stereospecific hDA reaction is, in fact the, reaction of transient quinomethide 26 with the humulene dienophile to form the dihydropyran ring. 

Maybe it could be rephrase as: hDA enzyme dehydrates the tropolone alchohol 25 to form a transient quinomethide 26 prior to the stereospecific hDA reaction with the humulene dienophile to form a dihydropyran.

Page 6 

Line152. "eupenifeldin 2" instead of "eupenifeldin"

Page 7 

Scheme 4. Use color dots to mark source of each compounds as Schemes 2 and 3. 

Page 8

Line 214.  "tropolone" instead of "topolone"

Line 220.  "tropolone" instead of "topolone"

Line 233-235. "Although compounds in this series have been isolated from various sources, it seems highly unlikley that BGCs have evolved that specify the synthesis of enantiomeric compounds and it is safe to assume that all compounds in this series should be 6S,7S"

Although the statement seems right in light of the cited data, enantiomeric antipods do exist in nature, for recent reviews of the occurrence of natural enantiomeric compounds (cases were both enantiomers exists in nature) 

* JM Finefield, DH Sherman, M Kreitman, RM Williams, Enantiomeric Natural Products: Occurrence and Biogenesis, Angew Chem Int Ed Engl.2012 May 14;51(20):4802–4836.doi:10.1002/anie.201107204.

** J-H Yu, Z-P Yu, RJ Capon,H Zhang, Natural Enantiomers: Occurrence, Biogenesis and Biological Properties, Molecules 2022, 27, 1279. https://doi.org/10.3390/molecules27041279

Then, it cannot be assumed safely that all compounds are 6S,7S without proper characterization of each compound.

Lines 239-241. "Since, in both cases, the 4-hydroxyl is also reported to be inverted relative to the C-4 hydroxyl of eupenifeldin 2, it is likely that ramiferin 10 and pughiinin A 11 should be reported as their respective enantiomers (corrected structures shown in Scheme 2)". 

Again, structures cannot be corrected without experimental evidence.  This should be clarified. Perhaps, it would be better to use the term "proposed structure"

Page 9. 

Line 243-2434 "Sarlah's recent synthetic work has proven the Western anti ring junction to be 6S,7S as mentioned before. This showed that the originally reported structures were enantiomeric to the true structures"

This could only be applied to structures of pycnidione 1 and epolone B 3

Line 245. "a strong" instead of "the good" nOe (or NOE correlations). Interpretation of NOE correlations are made with an adequate model to show which hydrogens are close. It should be pointed out which hydrogens are involved in the correlation with a model. 

Line 246. "supported" or "in agreement" instead of "confirmed"

Line 247. "(-)-epolone B" instead of "epolone B", "definitively" instead of "definativelty"

Line 249-252. "Since all later structures have been assigned on the basis of the original enantiomeric structure, it seems that all of the pycnidione family should be reassigned accordingly, including epolones A 6, B 3 and C 9[2] and dehydroxypycnidione 7.[2] Corrected structures are shown in Scheme 3." 

This could not be done without experimental evidence or reinterpretation of available spectroscopic data. Perhaps, it would be better to use the term "proposed structure" instead of "corrected structure"

Figure 5."tropolone" instead of "topolone"

Line 255-257.Phomeroids A and B (18 and 19) are reported to have an inverted secondary alcohol at C-4 (4R) relative to their proposed precursor eupenifeldin 2 (i.e. 4S). Since both compounds co-occur with eupenifeldin 2 in Phomopsis tersa it seems likely that C-4 should be redefined as 4S in these compounds.

In reference [13] Phomeroids A and B were described correctly, their absolute stereochemistry were determined by comparison of experimental and TDDFT calculated ECD. For eupenifeldin, only 1H and 13C were recorded in methanol-d4. 

Then, structures described in ref [13] cannot be corrected without further evidence or discussion based on the available data.     

Line 258-260. "This is emphasised by the fact that related compounds phomanolides A-F (14, 32, 35,15, 33, 17) are all assigned as 4S and all also co-occur with eupenifeldin 2"

In ref[10] phomanolides C-F were described correctly, their absolute stereochemistry were determined by comparison of experimental and TDDFT calculated ECD with a very similar calculation method of ref[13](phomeroids work). For eupenifeldin, 1H, 13C, and ECD were recorded and compared with simulated ECD spectra.

Then, the stereochemistry of phomanolides in refs[10-11] are in good agreement with eupenifeldin, but doesn´t necessarilly corresponds with phomeroids of ref[13]. A proper revision of spectroscopic evidence is needed in order to present corrected structures.   

Line 260-268. In order to establish an empiric rule for these compounds is important to present the complete set of spectroscopical data, with the solvent used to aqcuire each spectra. Macrocycles are difficult to analize and establish empirical rules, mainly because of its conformational freedom.

Line 273. Again, a proper characterization of each compound should be made, or use the term "proposed structure" intead of "corrected structure"

Line 274. "tropolone" instead of "topolone"

Line 280-283. The spectroscopical data of ref[9] should be compared with ref[1] and presented, in case that a structural revision is being made. In ref[9] I couldn´t find the spectroscopical data of "pycnidione or eupenifeldin", maybe there is a supplementary material but I couldn´t access.

Page 10 

Line 290."tropolone" instead of "topolone"

Line 295. "analogous stereoselectivity" instead of "identical stereoselectivity"

Line 307. "tropolone" instead of "topolone"

Page 11

Line 327."tropolone" instead of "topolone"

Reviewer 3 Report

This review will be of high significance and will attract readers interested in this field. I reviewed the manuscript in details and several times. I would highly recommend the publication of this manuscript in its current form. The manuscript is well written and prepared. 

Round 2

Reviewer 2 Report

I agree with all corrections, I added few minor comments. 

Page 1, Line 11(abstract) "stereochemical mis-assignments to be revised" instead of "stereochemical mis-assignments to be corrected"

Page 3, line 86. first instead of fist

Page 9, line 251. For pycnidione, H-4(beta face) and H-2(alpha face) are located in opposite sides of the 11-membered ring, so it isn´t complete clear the correlation between this nOe signal and the stereochemistry in C-4. Perhaps if this was already discussed in the original paper it will be sufficient to add the reference in the text.  

Page 10, line 280 to 284. Seems that type of letter has been changed

Page 10, line 281. "a nOe correlation" instead of "an nOe correlation" 

Author Response

Thanks for the additional comments. I made the minor text corrections.

For pycnidione the H-2,H-4 nOe was observed in Cai et al J. Nat prod. 1998. I have given the citation, but also changed the text to state 'observed nOe' rather than 'strong nOe' as it says in the paper itself. In fact, the distance in the crystal structure between these protons is 3.2 A, so an nOe is not inconceivable. If the 4-stereochemistry were reversed the distance would be 4.2 A, making an observation of an nOe unlikely.